# Structure of human Frizzled5 by fiducial-assisted cryo-EM supports a heterodimeric mechanism of canonical Wnt signaling

Naotaka Tsutsumi[1,2,3], Somnath Mukherjee[4], Deepa Waghray[1], Claudia Y Janda[1,2,5], Kevin M Jude[1,2,3], Yi Miao[1,2], John S Burg[1,2], Nanda Gowtham Aduri[2,6], Anthony A Kossiakoff[4], Cornelius Gati[2,6]*, K Christopher Garcia[1,2,3]*

[1]Department of Molecular and Cellular Physiology, Stanford University School of Medicine, Stanford, United States; [2]Department of Structural Biology, Stanford University School of Medicine, Stanford, United States; [3]Howard Hughes Medical Institute, Stanford University School of Medicine, Stanford, United States; [4]Department of Biochemistry and Molecular Biology, The University of Chicago, Chicago, United States; [5]Princess Máxima Center for Pediatric Oncology, Utrecht, Netherlands; [6]SLAC National Accelerator Laboratory, Bioscience Division, Menlo Park, United States

**Abstract** Frizzleds (Fzd) are the primary receptors for Wnt morphogens, which are essential regulators of stem cell biology, yet the structural basis of Wnt signaling through Fzd remains poorly understood. Here we report the structure of an unliganded human Fzd5 determined by single-particle cryo-EM at 3.7 Å resolution, with the aid of an antibody chaperone acting as a fiducial marker. We also analyzed the topology of low-resolution XWnt8/Fzd5 complex particles, which revealed extreme flexibility between the Wnt/Fzd-CRD and the Fzd-TM regions. Analysis of Wnt/β-catenin signaling in response to Wnt3a versus a 'surrogate agonist' that cross-links Fzd to LRP6, revealed identical structure-activity relationships. Thus, canonical Wnt/β-catenin signaling appears to be principally reliant on ligand-induced Fzd/LRP6 heterodimerization, versus the allosteric mechanisms seen in structurally analogous class A G protein-coupled receptors, and Smoothened. These findings deepen our mechanistic understanding of Wnt signal transduction, and have implications for harnessing Wnt agonism in regenerative medicine.

*For correspondence:
cgati@stanford.edu (CG);
kcgarcia@stanford.edu (KCG)

## Introduction

Frizzled (Fzd) receptors recognize lipidated and glycosylated growth factors, called Wnts, that play crucial roles in stem cell biology, embryonic development and adult tissue regeneration (*Nusse and Clevers, 2017*). Although Fzd receptors are part of the class F G protein-coupled receptor (GPCR) family and involved in G protein signaling (*Wright et al., 2018*), they primarily activate the canonical β-catenin pathway (*Carmon and Loose, 2008*; *He et al., 1997*). Binding of Wnt to Fzd induces formation of a ternary signaling complex with the co-receptor LRP5/6, recruiting the intracellular complex comprised of Dishevelled (Dvl), Axin, APC, CK1 and GSK3. This event silences the destruction complex, leading to accumulation of transcriptional co-activator β-catenin and the induction of Wnt target genes. Because aberrant activation of Wnt/β-catenin signaling can lead to tumorigenesis, inhibition of Wnt signaling is a prime target as a cancer therapy (*Anastas and Moon, 2013*).

Furthermore, Wnt's essential function in orchestrating stem cell proliferation and differentiation have also prompted interest in its use to promote tissue regeneration upon injury (*Clevers et al., 2014*).

Unlike class A GPCRs, where ligand-induced allostery is well established, the structural and mechanistic basis of Wnt signaling through Fzd remains enigmatic. Wnt signaling requires both a seven transmembrane (TM) GPCR-like (Fzd) and a 1TM (LRP5/6) receptor. On one hand, the clear similarity of Fzd to GPCRs suggests that ligand-induced conformational change could be at play (*Wright et al., 2018*). On the other hand, canonical Wnt/β-catenin signaling requires Fzd cross-linking with LRP5/6 by both Wnts and Norrin, implying a heterodimerization mechanism analogous to single-pass TM receptors such as cytokine receptors (*Janda et al., 2017*). To what extent Wnt-induced conformational changes versus simple Fzd/LRP6 heterodimerization play a deterministic role in signaling is not completely clear. Moreover, although structural information on the Fzd cysteine-rich domain (CRD) and its recognition mechanism with both Wnts and Norrin (*Chang et al., 2015*; *Hirai et al., 2019*; *Janda et al., 2012*) is available, a structural understanding of the Fzd-7TM regions has lagged behind (*Zhang et al., 2018*).

The class F GPCRs include Smoothened (Smo) and ten Fzd family members which are classified into five subgroups based on homology and Wnt specificity: Fzd1/2/7, Fzd3/6, Fzd4, Fzd5/8 and Fzd9/10 (*MacDonald and He, 2012*). The overall architecture of Fzd receptors consists of an N-terminal CRD, or ligand-binding domain, that serves as the Wnt binding module, and a seven-transmembrane domain (7TM) that is connected to the CRD by a long hinge region (*Zhang et al., 2018*; *Figure 1—figure supplement 1*). The hinge region can be further subdivided into a flexible segment from the CRD to the first conserved cysteine residue, and the ordered segment from the cysteine to TM1, with the total lengths varying from 50 to 115 amino acid residues. The hFzd4-7TM structure, lacking the CRD, has been recently reported along with structure-based mutagenesis data and molecular dynamics simulations (*Yang et al., 2018*), but it contains thermostabilized mutations that make it difficult to ascertain the relevance of the TM conformations in comparison to Smo and class A GPCRs.

In order to deduce insights into the structural basis of Fzd activation, we determined a fiducial-assisted cryo-EM structure at 3.7 Å resolution of hFzd5, lacking any thermostabilizing mutations. We also analyzed Wnt/Fzd complex particles that revealed a flexibly connected Wnt/CRD module that does not allow us to resolve a high-resolution structure of a stable binary Wnt/Fzd complex. The Fzd-TM structure guided the design of a series of mutations to explore the role of transmembrane coupling in signaling by both Wnt and a surrogate agonist. The identical structure-activity relationships shown by the natural versus the synthetic bi-specific agonist strongly suggest that Wnt/β-catenin signaling through Fzd is largely dependent on cross-linking with LRP6, rather than class-A GPCR-like conformational changes. Thus, the canonical Wnt/β-catenin signaling mechanism is more reminiscent of ligand-induced heterodimerization of single-pass receptors, such as cytokine and receptor Tyrosine kinases (RTK). Understanding this signaling mechanism is key to utilizing the canonical Wnt/β-catenin pathway in regenerative medicine.

## Results and discussion

### Structure determination and analysis

The initial attempts at reconstructing the 65 kDa hFzd5 molecule were hampered due to the particle shape being dominated by the spherical features of the detergent micelle, an issue that complicates structure determination of small membrane proteins by cryo-EM. To facilitate unambiguous alignment of the particles, we sought to introduce features into the particle that would both increase its size and provide an asymmetric shape that would enable facile alignment for 3D reconstruction of hFzd5. We introduced the BRIL domain (cytochrome b562 RIL) fused in place of intracellular loop 3 of hFzd5. In order to optimize a rigid connection between hFzd5 and BRIL, we grafted two short linkers, each comprising five amino acid residues from $A_{2A}$ adenosine receptor-TMs between TM5/6 of Fzd5 and BRIL (See Methods), as used for the first full-length Smo structure (*Byrne et al., 2016*). We employed an anti-BRIL Fab, BAG2 (*Mukherjee et al., 2020*; *Figure 1a*) with an anti-Fab nanobody (Nb) (*Ereño-Orbea et al., 2018*) to complex with the BRIL fusion hFzd5 (hFzd5$_{ICL3}$BRIL). The BRIL/Fab/Nb module was designed as a 'fiducial' marker for particle image alignment (*Mukherjee et al., 2020*). We purified hFzd5$_{ICL3}$BRIL from 293S GnTI$^-$ cells in a glycol-diosgenin/

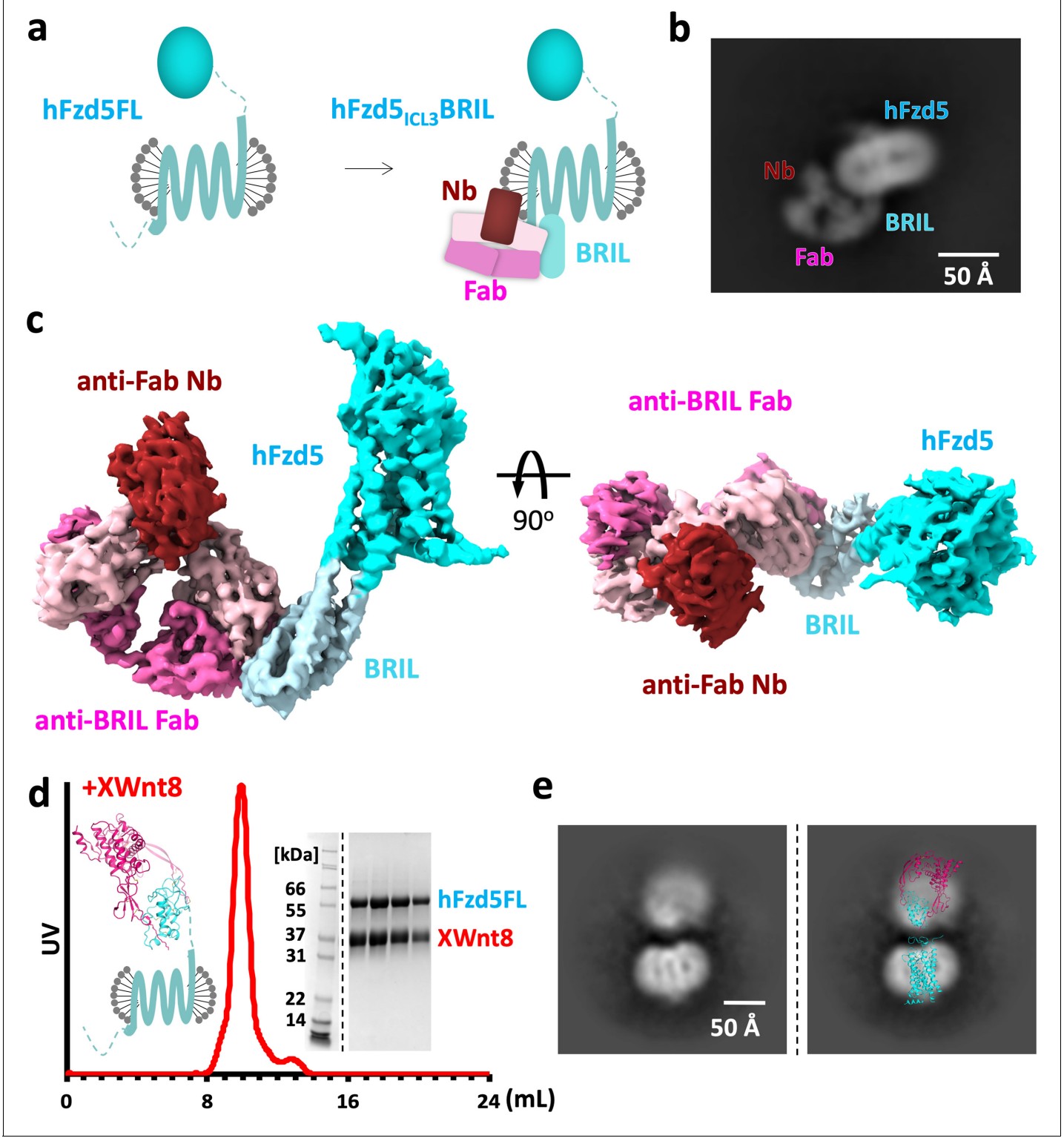

**Figure 1.** Design scheme and cryo-EM analysis of hFzd5. (**a**) Cartoon representation of the strategy for hFzd5 particle decoration by anti-BRIL Fab and anti-Fab Nb. These chaperones double hFzd5 molecular weight and render it asymmetric in detergent micelle, thus working as a fiducial maker for image alignment. (**b**) A selected 2D class average showing the side view of monomeric hFzd5$_{ICL3}$BRIL/Fab/Nb. (**c**) Overall EM volume around the structural model of hFzd5 (cyan), BRIL and the linker between BRIL and Fzd5 (light blue), Fab heavy chain (pink), Fab light chain (light pink) and Nb (wine). (**d**) The size-exclusion chromatography profile, SDS-PAGE of peak fractions, and (**e**) the selected 2D class average of XWnt8/hFzd5FL. The full

*Figure 1 continued on next page*

*Figure 1 continued*

SDS-PAGE and 2D classes are shown in *Figure 1—figure supplement 5*. Models of XWnt8/mouse Fzd8CRD (mFzd8CRD) (PDB ID: 4F0A) and hFzd5 are overlaid on the blobs to show the relative size of densities.

The online version of this article includes the following figure supplement(s) for figure 1:

**Figure supplement 1.** Multiple sequence alignment of human class F GPCRs.
**Figure supplement 2.** Purification, data collection and 2D classification of hFzd5$_{ICL3}$BRIL/Fab/Nb.
**Figure supplement 3.** Cryo-EM data analysis of hFzd5$_{ICL3}$BRIL/Fab/Nb.
**Figure supplement 4.** Cryo-EM map of representative built in model.
**Figure supplement 5.** Purification, data collection and 2D class averages of the XWnt8/hFzd5FL complex.

cholesterol hemisuccinate (GDN/CHS) micelle and incubated it with excess Fab and Nb before the size-exclusion chromatography step (*Figure 1—figure supplement 2a*). The peak fraction of the complex was concentrated to 8 mg/ml with digitonin as an additive for single-particle cryo-EM analysis (*Figure 1—figure supplements 2b,c* and *3*).

The vast majority of 2D class averages of the 'fiducial-assisted' Fzd5 were monomeric particles with the Fab/Nb module clearly discernable, but with sparse density for the Fzd-CRD (*Figure 1b* and *Figure 1—figure supplement 2c*). It has been suggested that Fzd receptors are capable of forming homodimers in both Wnt-dependent and -independent manners (*Carron et al., 2003*; *Hirai et al., 2019*; *Nile et al., 2017*). We observed a small population (~5% of the monodisperse particles) of parallel hFzd5 dimers in the 2D class averages (*Figure 1—figure supplement 2c*, marked with red boxes), which places the ICL3BRIL/Fab/Nb units on opposite sides of the dimeric receptor. The relative position of two Fabs indicates the TM4s are likely at the dimer interface given the expected GPCR fold, consistent with the crystal structure of hFzd4TM (*Yang et al., 2018*). Since Fzd appears to be monomeric on the surface of live cells, and hetero-trimerization of Wnt/Fzd with LRP6 is sufficient for activation of the intracellular β-catenin signaling (*Janda et al., 2017*), the role of observed hFzd5 dimers in signaling remains unclear (*Tao et al., 2019*).

3D classification of selected monomeric particles indicated the fusion between the TM region and BRIL was not completely rigid (*Figure 1—figure supplement 3a*). Two particle classes with clear TM density and well-defined features of the Fab/Nb module showed slightly varied angles (approximate median angular variation of 2°). We initially focused on the class with the highest resolution and most clearly resolved extracellular features from the final heterogeneous refinement step, yielding a 3D map with a nominal resolution of 4.6 Å at the Fourier shell correlation (FSC) criterion of 0.143. To overcome the residual flexibility between hFzd5 and BRIL, local non-uniform refinement was performed in cryoSPARC (*Punjani et al., 2017*) with a fulcrum on the hFzd5-BRIL linker. The final 3D map was created by combining the two different classes, resulting in a 3.7 Å map of monomeric hFzd5$_{ICL3}$BRIL/Fab/Nb with substantially better TM features allowing model building (*Figure 1c*, *Table 1* and *Figure 1—figure supplements 3* and *4*).

The EM volume is defined for the extracellular loops (ECL) 1–2 and the hinge domain (C192-F227). However, the CRD, the CRD-proximal flexible hinge and a part of ECL3 (A27-Y150, N151-K191 and P487-P498, respectively) are either missing or ambiguous in the 3D reconstruction, suggesting intrinsic flexibility between the 7TM and the ligand binding-module. Given the flexibility between the CRD and 7TM regions and their lack of intimate structural coupling, it seems unlikely that Wnt binding to CRD domain could be propagated through a conformational change to the 7TM bundle to modulate signaling in a manner analogous to authentic GPCRs. Indeed, the hinge of Fzd8, a Fzd5/8 subfamily member, is also quite extended and is ~40 amino acid residues longer than the one of Fzd5, but they share similar Wnt specificity and signaling properties (*Voloshanenko et al., 2017*).

We also imaged *Xenopus* Wnt8 (XWnt8) bound to full-length hFzd5 (hFzd5FL) without the BRIL fusion, the Fab, or the Nb (*Figure 1d,e* and *Figure 1—figure supplement 5*). After successfully purifying a stable complex, 2D class averages revealed two rather blurred densities that match the size of Wnt/Fz in detergent micelle. However, the densities for both regions are limited to low resolution due to extreme segmental variation and flexibility, and therefore do not allow us to resolve a complete structure of the complex. It is likely that LRP6, missing in our complex, could stabilize the Wnt/CRD module. Nevertheless, an approximate placement of hFzd5FL and XWnt8 in the 2D class averages reveals Wnt projecting straight up from the Fzd-TM-micelle, with a narrow stricture of missing

**Table 1.** Cryo-EM data collection and refinement statistics

| | hFzd5$_{ICL3}$BRIL/anti-BRIL Fab/anti-Fab Nb (EMD-21927) (PDB ID: 6WW2) |
| --- | --- |
| Data collection and processing | |
| Magnification | 81,000x |
| Voltage (kV) | 300 |
| Electron exposure (e/Å$^2$) | 50 |
| Defocus range (μm) | −0.8 to −2.0 |
| Pixel size (Å) | 1.078 |
| Symmetry imposed | C1 |
| Initial particle images (no.) | 6,483,398 |
| Final particle images (no.) | 369,704 |
| Map resolution (Å) FSC threshold | 3.70 0.143 |
| Map resolution range (Å) | 3.2 to 5.9 |
| Refinement | |
| Initial model used (PDB code) | 5L7D, 6BD4, 6ANI, 6CBV |
| Model resolution (Å) FSC threshold | 3.7/4.1 0.143/0.5 |
| Map sharpening $B$ factor (Å$^2$) | −124.4 |
| Model composition Non-hydrogen atoms Protein residues | 7,594 982 |
| $B$ factors (Å$^2$) Protein | 89.59 |
| R.m.s. deviations Bond lengths (Å) Bond angles (°) | 0.010 1.083 |
| Validation MolProbity score Clashscore Poor rotamers (%) | 1.87 5.73 0 |
| Ramachandran plot Favored (%) Allowed (%) Disallowed (%) | 89.59 10.31 0.10 |

density for the linker region. These 2D classes represent an average relative position of the two molecules that exist in a spectrum of angular variations. This allowed us to visualize both Wnt and Fzd densities, but probably represent only a part of many alternative Wnt/Fzd-CRD positions in the total population of particles. Attempts of 3D classification and reconstruction were not successful due to their severe segmental heterogeneity. Despite the limitations of the Wnt/Fzd data, the particles further support the conclusion that the Wnt/Fzd-CRD module is not in intimate contact with the TM-regions of Fzd and is therefore unlikely to participate in a tightly structurally coupled allosteric relay upon Wnt binding.

Although Smo has one of the shortest hinges among human Class F GPCRs, the cryo-EM structure of the Smo/G protein complex showed the EM map of the CRD was not well-defined (*Qi et al., 2019*). Nevertheless, the crystal structures of full-length Smo showed that the CRD could 'sit' on the ordered extracellular loops (*Byrne et al., 2016*) suggesting Smo-CRD is less flexible than Fzd-CRD. The freedom of the Fzd-CRD should enable Fzd to 'search' for ligands encountered in the membrane from the side of the receptor, not just from the top.

## Structural comparison between hFzd5, hFzd4 and hSmo

Human Fzd5 belongs to a distinct subfamily from hFzd4, but they share high similarity with the 7TM GPCR fold and have ~62% sequence identity in TMs (excluding the extracellular TM6-ECL3 residues 6.58–6.77, generic numbering for class F GPCRs *Isberg et al., 2015*). In the Fzd5 structure, the ECL2 forms a lid comprised of β-sheets and loops (*Figure 2a,b*). The hinge region forms an ordered structure that, along with ECLs, occludes the orthosteric ligand binding pocket defined for Smo and class A GPCRs. In contrast to the lipid GPCRs (EP3, EP4, and TP) that possess complete ECL2 lids with a gap between TMs (*Audet et al., 2019*; *Fan et al., 2019*; *Morimoto et al., 2019*; *Toyoda et al., 2019*), neither the Fzd5 nor Fzd4 structure shows a large opening within the TMs, implying the receptors' pocket would likely occlude small molecules. There are pronounced differences in the structure of the extracellular region of Fzd5, compared to the ECL structures of Fzd4 and Smo (*Figure 2c*). The hinge and linker domain of Fzd5 are more extended than those of Fzd4 and Smo, which are the shortest among the Class F GPCRs. The hinge domains of Fzd5/8 contain five cysteine residues, in contrast to three cysteines for the other subclasses in the Fzd/Smo family. Three cysteine residues in the flexible hinge and the start of the hinge domain are unmodeled in the structure.

Compared with the extremely short ECL3 of Fzd4 comprising a small α-helix sitting parallel to the membrane, Fzd5 has the longer ECL3, annotated as a part of TM6 by sequence similarity with Smo. However, compared to Smo, Fzd5-ECL3 forms a shorter α-helix with an extended flexible loop that is not clearly defined in the EM map (*Figure 2c* and *Figure 1—figure supplement 4c*), which might favor ligand-induced allostery in Smo that is not seen in Fzd. For example, it has been proposed that during Smo signaling, the long TM6-ECL3 α-helix interacts with the CRD to sense ligand-induced conformational rearrangement by the CRD. This could facilitate allosteric communication between the CRD, TM6 and TM7 for receptor activation (*Deshpande et al., 2019*; *Huang et al., 2018*). Fzd5/8-ECL3 has a CXC motif that forms either a rare intra-loop disulfide bond or two disulfide bonds with the unique orphan cysteines in the flexible hinge (*Figure 1—figure supplement 1*). Smo and most Fzds have two cysteines dispersed in their ECL3s, and Smo shows an intra-ECL3 disulfide bond in the structure. In contrast, there is no ECL3 cysteine for Fzd4. The differences in cysteine distribution in the hinge region and ECL3 between Fzd4 and Fzd5 might explain the cell signaling data from a chimeric Fzd4, where the whole hinge region was replaced by the Fzd5 hinge (*Bang et al., 2018*). Although cell-surface expression was confirmed, the chimeric receptor was completely inactive to Norrin stimulation, and it was concluded that the loss of the Norrin-Fzd4 hinge interaction disrupted signaling. However, a more plausible explanation may be that the disruption of the disulfide network impaired the structural integrity of the chimeric receptor. Consistent with this explanation, several Fzd4 mutations of the hinge cysteines (C181R, C204Y, and C204R) render Fzd4 inactive in patients with familial exudative vitreoretinopathy (*Milhem et al., 2014*).

In Fzd-7TM, the 'orthosteric' pocket for small-molecules (eg. vismodegib, Anta XV, TC114, LY2940680 and SAG1.3 for Smo-7TM) has a constriction that restricts ligand access (*Figure 2d*). In the putative 'druggable' pocket, Fzd5 has a bulky ionic residue K506$^{7.41}$ (superscript denotes generic numbering for class F GPCRs residue). This residue is conserved in ten Frizzled members, potentially restricting the small molecule access together with the neighboring bulky sidechain of Y507$^{7.42}$. The approximate minimum diameters of the pockets are 4.9 Å for Fzd5 and 4.2 Å for Fzd4, whereas Smo has a 5.8 Å diameter stricture near the entrance of the cavity. Alanine substitution of either K506$^{7.41}$ or Y507$^{7.42}$ severely impaired cell surface protein expression of full-length hFzd5, implying the importance of these residues for protein stability likely through inter-helical packing (*Figure 2—figure supplement 1*). While the small pockets evident in the Fzd-7TM structures might help to explain the difficulty of developing small molecule modulators for Wnt/β-catenin signaling, a recent report showed the Smo agonist SAG1.3 acted as a partial agonist for Fzd6-Gi signaling (*Kozielewicz et al., 2020*). Thus, small molecule signal modulation of Fzd remains an area of active investigation.

We also compared the cytoplasmic regions of the 7TM bundles of Fzd5, Fzd4 and Smo, which are the regions that interact with signaling adaptor proteins. TM6 plays an important regulatory role in the activation of class A GPCRs, and Smoothened in the context of G protein signaling, and is found in a similar position in both Fzd4 and Fzd5. However, the cytoplasmic end of Fzd5-TM5 is in an inward position compared to Fzd4-TM5 and aligns well with the Smo-TM5 in its inactive state (*Figure 3a*). In this region, Fzd5 and Smo have a phenylalanine residue in TM5 or TM6 that interacts

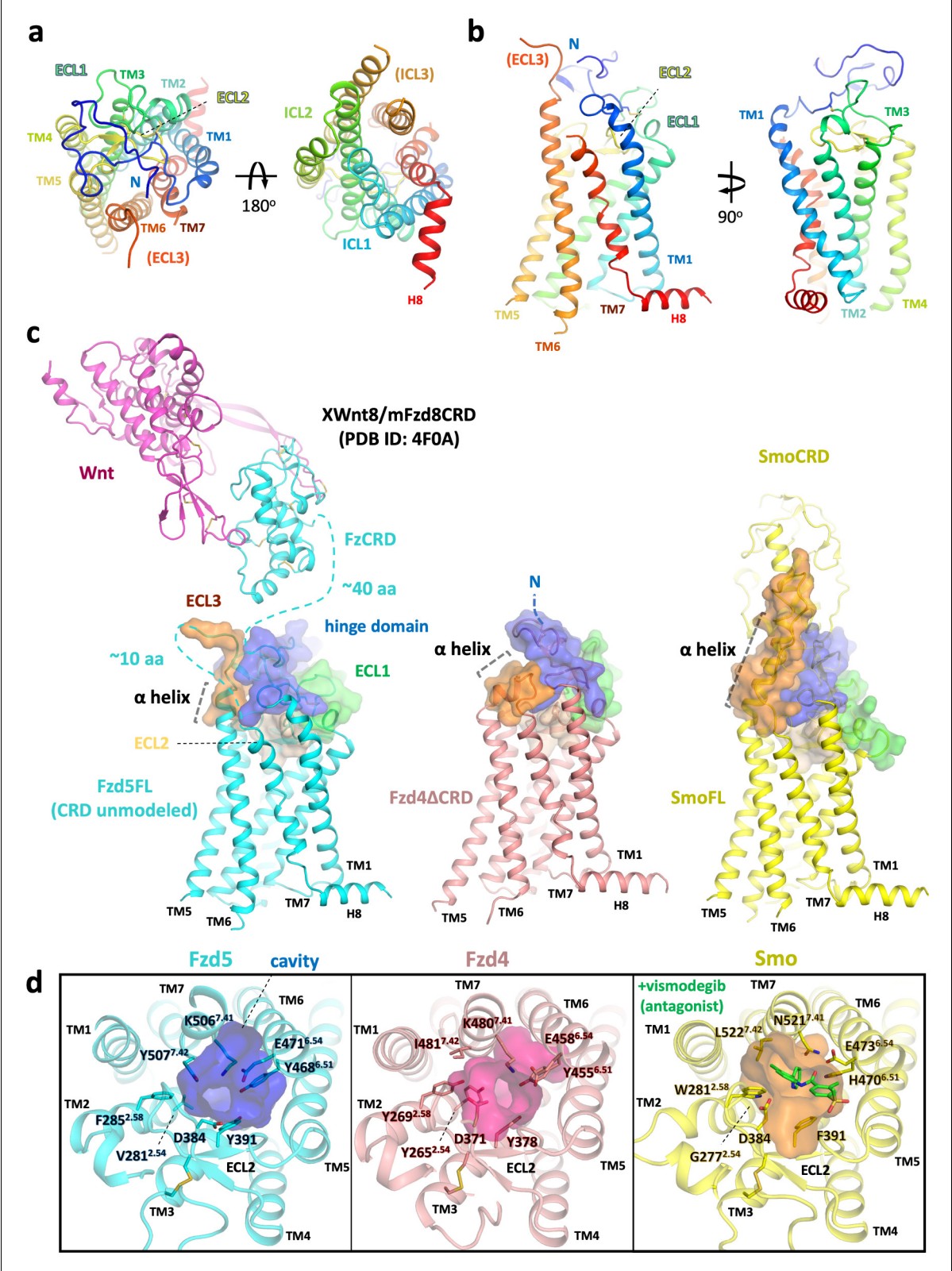

**Figure 2.** Overall structure of hFzd5 and structural comparison with hFzd4 and hSmo. (a) Top (left) and bottom (right) views, and (b) side views of hFzd5 colored by rainbow with blue on the N-terminus and red on the C-terminus of the structural model. (c) Comparison of the extracellular regions between hFzd5, hFzd4 (pink, PDB ID: 6BD4) and hSmo (yellow, PDB ID: 5L7D). Transparent surfaces are overlaid on the cartoon models and colored separately by the hinge domain (blue), ECL1 (green), ELC2 (wheat) and ELC3 (orange). XWnt8/mFzd8CRD structure (PDB ID: 4F0A) is displayed and connected to

*Figure 2 continued on next page*

*Figure 2 continued*

the hFzd5 structure by dashed line, to show the size of the LRP6 binding module relative to 7TM. The α-helices at TM6-ECL3 are indicated with gray dashed lines. (d) Top views of hFzd5, hFzd4, and hSmo around the Smo-7TM ligand binding site with potential gatekeeping residues. The cavity volume calculated using the CavityPlus server (*Xu et al., 2018*) are displayed as transparent surface representations, with a focus on the constrictions made by bulky resides of Fzd at the upper-core site. The antagonist vismodegib (from PDB ID: 5L7I) is overlaid on the hSmo structure in its inactive and *apo* state (PDB ID: 5L7D).

The online version of this article includes the following source data and figure supplement(s) for figure 2:

**Figure supplement 1.** Confirmation of hFzd5 receptor expression in HEK293T transfected cells.

**Figure supplement 1—source data 1.** Raw FACS histogram data for *Figure 2—figure supplement 1* showing cell-surface expression of hFzd5 variants.

with $W^{3.50}$. Additionally, there is a small aliphatic residue on the counterpart TM6 or TM5, respectively that apparently stabilizes the 'inward' conformation. Fzd4 has $L410^{5.66}$ and $V442^{6.38}$ at the corresponding positions that might result in looser packing between TM5-TM6.

An important question that could be related to receptor activation pertains to conformational changes of the Fzd-TM. For Fzd4, small distortions of the conformations of TM4 and TM7, including two kinks in TM7, were noted after molecular dynamics simulations of the crystal structure, leading to conjecture that TM4 and TM7 transitioned between two conformations to modulate signaling (*Yang et al., 2018*). TM4s and TM7s of Fzd4 and Fzd5 are closely structurally aligned, although Fzd5 $W520^{7.55}$ forms a more extensive π-cation interaction with $R449^{6.32}$ ($W494^{7.55}$ and $K436^{6.32}$ in Fzd4) (*Figure 3a,b*). The $K/R^{6.32}$-$W^{7.55}$ interaction, known as the 'ionic lock' is a crucial molecular switch for intracellular signaling in class F GPCRs, and it has been suggested that the Dvl- versus G protein-biased states exist with the $K/R^{6.32}$-$W^{7.55}$ interaction either on or off, respectively (*Wright et al., 2019*). $R449A^{6.32}$ in 'canonical' Fzd5 abolished Dvl recruitment and Wnt3a-induced β-catenin signaling, while $R416A^{6.32}$ in 'non-canonical' Fzd6 increased G protein signaling (*Wright et al., 2019*). Fzd5-H8, which has another Dvl-binding KTxxxW motif, undergoes a helical half-turn outward transition compared with Fzd4 (*Figure 3c*). The KTxxxW motif is known to interact with the PDZ domain and the DEP domain of Dvl (*Gammons et al., 2016*; *Wong et al., 2003*), where point mutations of the conserved K, T and W residues impair the canonical β-catenin signaling.

## Probing the effect of structure-guided hFzd5 mutations on signaling

Does Wnt contain 'intrinsic' structural properties necessary to activate canonical signaling beyond simply acting as a Fzd/LRP6 cross-linker? Delineating between a Wnt-induced allosteric mechanism through Fzd, a la' GPCR, and a simple Fzd/LRP heterodimerization mechanism remains challenging. Thus, we approached this experiment by comparing structure-activity relationships for canonical signaling induced by the natural ligand Wnt3a, versus a synthetic agonist that cross-links Fzd5 and LRP6 (*Figure 4a,b*; *Janda et al., 2017*). 'Surrogate Wnt' molecules serve as cross-linkers of Fzd and LRP6 receptors to form signaling heterodimers, but with distinct binding modes compared to Wnt, and therefore are devoid of any 'intrinsic' structural features of Wnt that might be contributing to an allosteric mechanism. We examined β-catenin/T cell factor (TCF) signaling downstream of hFzd5 upon stimulation with either hWnt3a or a surrogate Wnt using the Fzd1/2/4/5/7/8 knockout cell line (*Voloshanenko et al., 2017*; *Figure 4* and *Figure 2—figure supplement 1*). We chose mutants from the Fzd5-TM structure that appear to mediate important inter-helical interactions that might be associated with allostery and have been previously noted in Fzd. $K506A^{7.41}$ and $Y507A^{7.42}$ mutation completely or severely impaired cell-surface Fzd expression, respectively, whereas other mutants showed comparable expression levels to the wild-type receptor (*Figure 2—figure supplement 1*). With wild-type Fzd5 expression, the cells signaled in response to Wnt3a and the surrogate Wnt in a concentration-dependent manner, confirming that the output signaling is Fzd5 specific (*Figure 4a, b*). We then tested a series of mutants in this assay system. Fzd5-$Y507A^{7.42}$ signaled as robustly as the wild-type receptor regardless of its low expression, indicating the expression levels are not limiting signaling strength. The ionic lock mutants Fzd5-$R449A^{6.32}$ and Fzd5-$W520A^{7.55}$ completely ablated or significantly reduced the signaling stimulated by both the agonists, in accordance with previous studies showing the importance of this interaction for signaling (*Wright et al., 2019*). $W522A^{7.55}$ and $T252F^{1.49}$/$T534F^{8.58}$ mutants did not alter signaling compared to wild-type Fzd5, while $W238A^{1.35}$ showed slightly lower signal compared to wild-type Fzd5 for both agonists. The

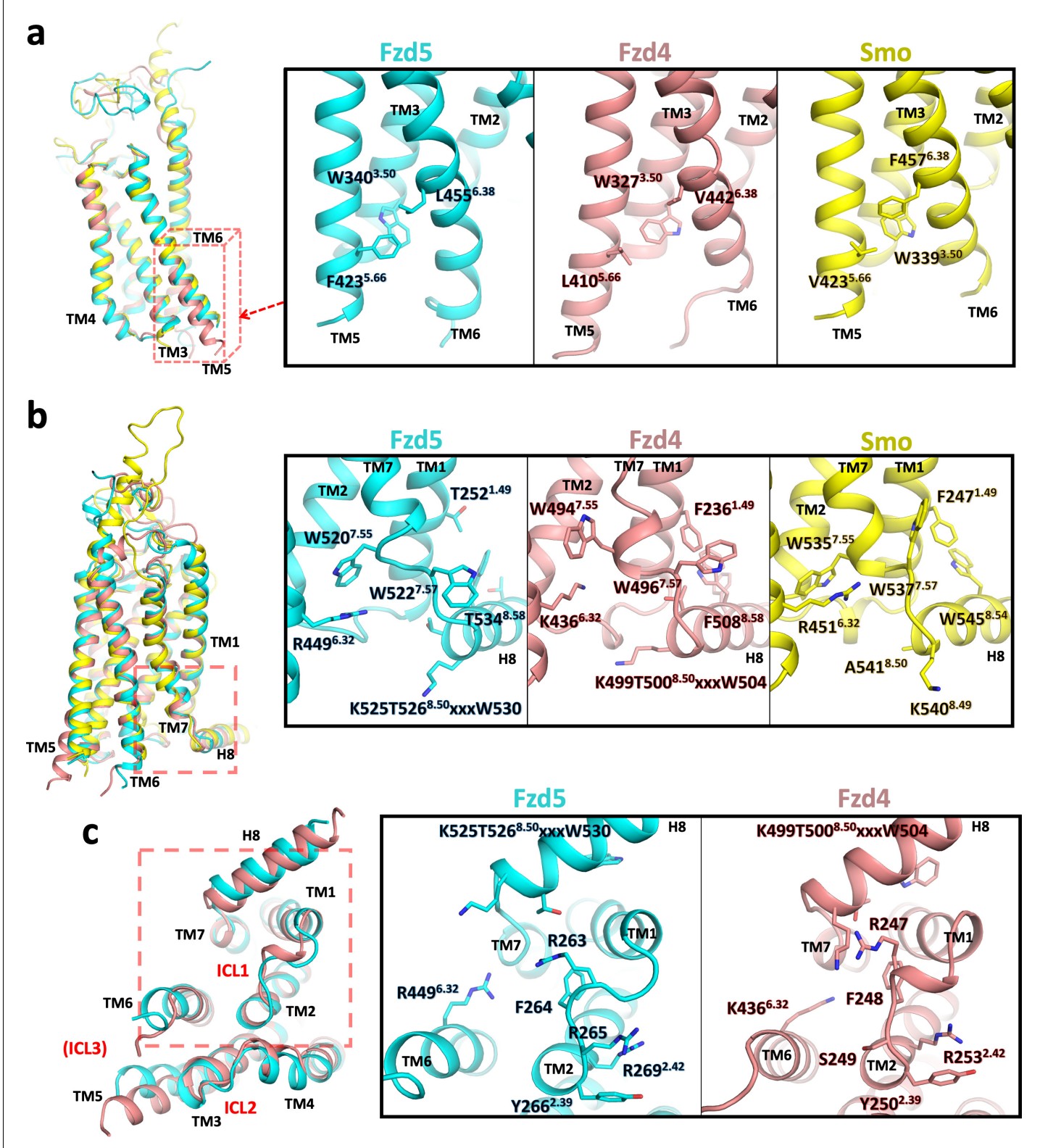

**Figure 3.** Conformational differences between hFzd5, hFzd4 and hSmo. (**a, b**) Structural comparison of hFzd5, hFzd4 and hSmo (**a**) at the cytoplasmic end of TM5, and (**b**) the K/R$^{6.32}$-W$^{7.55}$ ionic lock with the aromatic network around H8. (**c**) Bottom views of hFzd5 and hFzd4 showing structural rearrangement of ICL1 and H8 important for Dvl binding. Each molecule is colored by cyan (hFzd5), pink (hFzd4, PDB ID: 6BD4) or yellow (Smo, PDB ID: 5L7D).

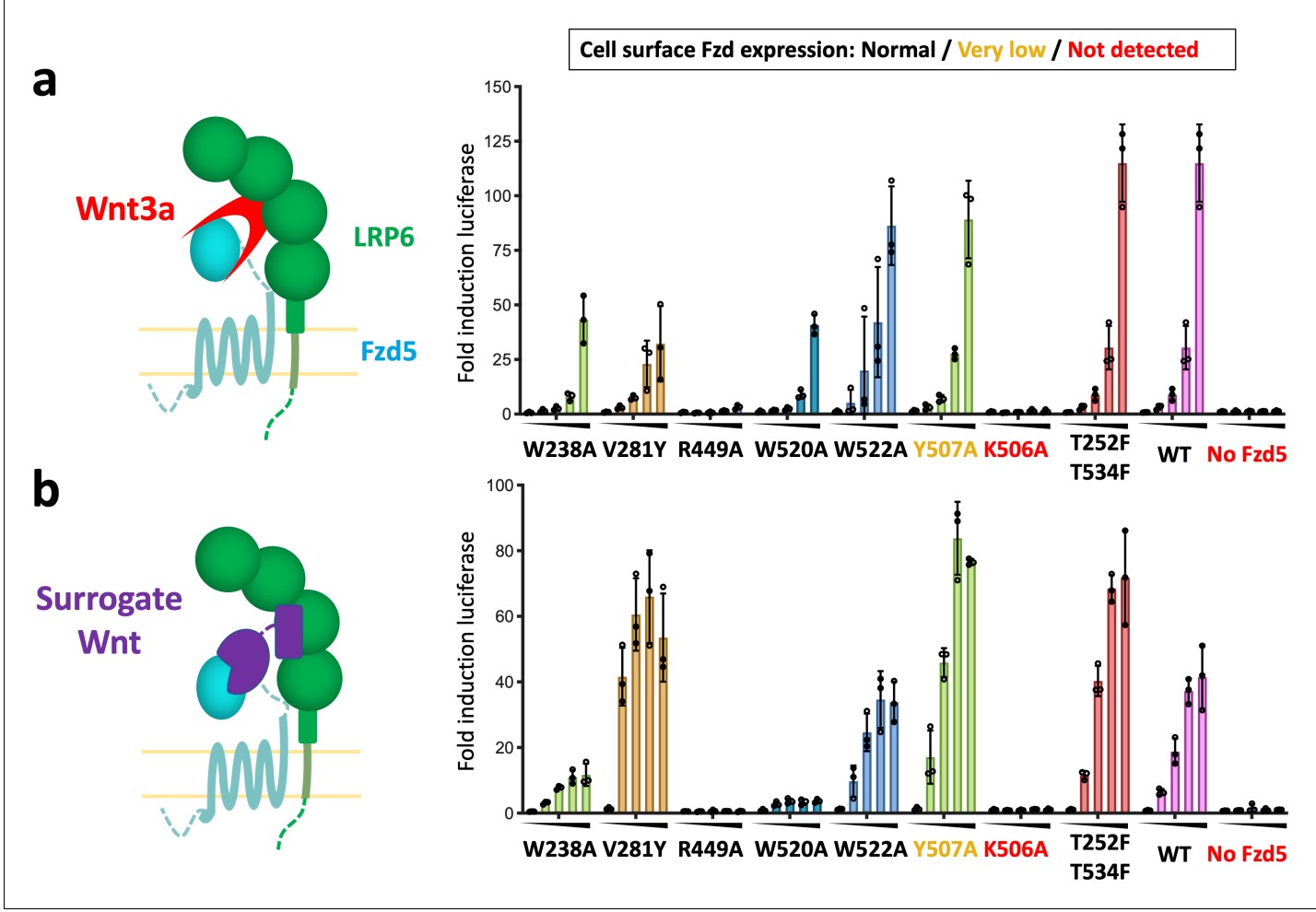

**Figure 4.** Structure-guided hFzd5 mutant signaling assays. Schematic presentation of the ternary complex formation, and hFzd5-mediated signaling upon stimulation by (a) Wnt3a and (b) surrogate Wnt agonist. Non-lipidated, water-soluble surrogate Wnt would form the complex in a different geometry from natural Wnt ligands. Each series was stimulated (a) by Wnt3a at the concentrations of 0, 1.6, 3.1, 6.3 or 12.5 nM, or (b) by surrogate Wnt at the concentrations of 0, 0.25, 1, 4 or 16 nM, displayed from left (no agonist) to right (the highest concentration) and indicated by black triangles. The bars and error bars represent means and standard deviations, respectively, of data points from three technical replicates shown as circles.

The online version of this article includes the following source data for figure 4:

**Source data 1.** Raw data points for the signaling experiments shown as circles in *Figure 4*.

V281Y[2.54] mutation, which is at the small molecule binding site in Smo, yielded a lower signal with Wnt3a stimulation, but a stronger signal for the surrogate Wnt than the wild type. In general, the response patterns were similar between Wnt3a and surrogate Wnt. In a previous mutational analysis of Fzd4 signaling by Wnt3a or Norrin, a generally similar signaling pattern was observed for both ligands (*Yang et al., 2018*).

Collectively these results strongly imply that Wnt does not initiate conformational changes in Fzd that are intrinsic to Wnt, and that Wnt principally acts as a cross-linker. This is further supported by the long and flexible linker between the Fzd-TM and CRD, which does not allow intimate structural coupling between ligand binding and conformational changes in the TM. This stands in contrast to Smo, which has a more rigid linker, allowing allosteric signal transduction by sensing the stimulus on CRD. It is possible that G protein signaling is more reliant on the conformational change of Fzds. Also, it is still unclear if the proximity of LRP6 to Fzd induces conformational changes in Fzd as a result of heterodimerization, but if so, this would be driven by structural properties intrinsic to the receptors, not Wnt. The canonical Wnt system, comprising both a 7TM GPCR-like receptor (Fzd) and a single-pass 1TM receptor (LRP5/6), has presented a mechanistic conundrum in that GPCRs signal

through ligand-induced allostery, whereas single-pass TM receptors generally signal via ligand-induced dimerization, exemplified by RTK and JAK/STAT cytokine receptors (*Wilmes et al., 2020*). In this case, the preponderance of structural and mechanistic data point to a unique example of a heterodimerization-driven mechanism by a GPCR-like receptor Fzd and a Type-I receptor LRP5/6. This feature opens up the possibility of 'tuning' Wnt signaling through the use of alternative agonist molecules, such as antibodies and other dimeric scaffolds (*Tao et al., 2019*), to obtain a more granular understanding of Wnt biology and for developing regenerative therapeutics.

# Materials and methods

## Key resources table

| Reagent type (species) or resource | Designation | Source or reference | Identifiers | Additional information |
|---|---|---|---|---|
| Cell line (*Spodoptera frugiperda*) | Sf9 | ATCC | CTL-1711 (RRID:CVCL_0549) | Insect cells used for baculovirus production |
| Cell line (*Homo sapiens*) | HEK293S GnTI⁻ | Gift from Prof. H Gobind Khorana (PMID:12370423) | | Mammalian cells used for baculovirus expression of hFzd5FL and hFzd5$_{ICL3}$BRIL |
| Cell line (*Homo sapiens*) | HEK293T with Fzd1/2/4/5/7/8 KO | Gift from Prof. Michael Boutros (PMID:28733458) | | For signaling assay |
| Cell line (*Mus musculus*) | Mouse monoclonal M1 hybridoma | Gift from Prof. Brian K Kobilka (PMID:17962520) | | To purify anti-FLAG M1 (mouse IgG2a) antibody to prepare FLAG affinity column (15 mg/ml resin). This was used for purification of hFzd5FL and hFzd5$_{ICL3}$BRIL. |
| Strain, strain background (*Escherichia coli*) | BL21(DE3) | New England Biolabs | C2527I | *E. coli* strain for expression of the nanobody |
| Strain, strain background (*Escherichia coli*) | BL21 (gold) | Agilent Technologies | 230130 | *E. coli* strain for expression of the Fabs |
| Transfected construct (*Homo sapiens*) | pRK5 hFzd5 wild-type | *Yu et al., 2012* (PMID:23095888) | | For signaling assay |
| Transfected construct (*Homo sapiens*) | pRK5 hFzd5 mutants | this study | | For signaling assay |
| Recombinant DNA reagent | 7xTCF-Ffluc | Addgene | 24308 | For signaling assay |
| Recombinant DNA reagent | BestBac Linearized Baculovirus DNA 2.0 | Expression Systems | 91–002 | For baculovirus production |
| Peptide, recombinant protein | Wnt3a | R and D Systems | 5036-WN | For signaling assay |
| Peptide, recombinant protein | Surrogate Wnt | Miao et al. *Cell Stem Cell* 2020 | | For signaling assay |

*Continued on next page*

*Continued*

| Reagent type (species) or resource | Designation | Source or reference | Identifiers | Additional information |
|---|---|---|---|---|
| Peptide, recombinant protein | DRPB_Fz8 | *Dang et al., 2019* (PMID:31086346) | | For signaling assay |
| Commercial assay or kit | Luciferase assay kit | Promega | E152A | For signaling assay |
| Chemical compound, drug | n-Dodecyl-β-D-Maltoside (DDM) | Anatrace | D310S | Membrane protein purification |
| Chemical compound, drug | Lauryl Maltose Neopentyl Glycol (LMNG) | Anatrace | NG310 | Membrane protein purification |
| Chemical compound, drug | Glyco-Diosgenin (GDN) | Anatrace | GDN101 | Membrane protein purification |
| Chemical compound, drug | Cholesterol Hemisuccinate tris Salt (CHS) | Anatrace | CH210 | Membrane protein purification |
| Chemical compound, drug | Digitonin | EMD Millipore | 300410 | Cryo-EM specimen freezing additive |
| Software, algorithm | DigitalMicrograph | Gatan | | Microscope alignment and cryo-EM data collection |
| Software, algorithm | serialEM | https://bio3d.colorado.edu/SerialEM/ (PMID:16182563) | version 3.6 | Cryo-EM data collection |
| Software, algorithm | MotionCor2 | https://emcore.ucsf.edu/ucsf-motioncor2/ (PMID:28250466) | | Motion correction of cryo-EM movies |
| Software, algorithm | cryoSPARC | https://cryosparc.com/ (PMID:28165473) | version 2.12.14 | Cryo-EM data processing |
| Software, algorithm | Coot | https://www2.mrc-lmb.cam.ac.uk/personal/pemsley/coot/ (PMID:20383002) | | Structure model building |
| Software, algorithm | Isolde | https://isolde.cimr.cam.ac.uk/ (PMID:29872003) | | Structure model building/refinement |
| Software, algorithm | Phenix suite | https://www.phenix-online.org/ (PMID:20124702) | | Structure refinement |
| Software, algorithm | UCSF Chimera | https://www.cgl.ucsf.edu/chimera/ (PMID:15264254) | | Initial homology model docking |
| Software, algorithm | UCSF ChimeraX | https://www.rbvi.ucsf.edu/chimerax/ (PMID:28710774) | | Structural visualization/figure preparation/Isolde execution |
| Software, algorithm | PyMol | Schrödinger | | Structural visualization/figure preparation |
| Software, algorithm | GraphPad Prism 7 | GraphPad | | analysis of signaling data |

*Continued on next page*

*Continued*

| Reagent type (species) or resource | Designation | Source or reference | Identifiers | Additional information |
|---|---|---|---|---|
| Other | CNBr-Activated Sepharose 4 Fast Flow | GE Healthcare | 17098101 | For preparation of anti-FLAG sepharose |
| Other | Lipofectamine 2000 | Invitrogen | 11668030 | Transfection reagent for signaling assay |

## Cell lines

Sf9 cells were obtained from ATCC (CTL-1711, RRID:CVCL_0549), HEK293S GnTI⁻ cells were provided by Prof. H Gobind Khorana, Fzd1/2/4/5/7/8 knockout HEK293T cells were provided by Prof. Michael Boutros and M1 hybridoma cells were provided by Prof. Brian K Kobilka.

Cell line authentication was guaranteed by the sources where the cells were obtained. For Sf9, ATCC does not indicate the authentication method. For HEK293S GnTI⁻ cells, clonal selection was performed by the provider based on their GnTI activity, and the authentication was regularly monitored by purified protein's sensitivity to Endo H enzyme. For Fzd1/2/4/5/7/8 knockout HEK293T cells, the original cell authentication was confirmed with NGS sequencing by the provider, and the authentication was regularly monitored by their response to Wnt3a as shown in this study (negative and positive controls). For M1 hybridoma, clonal selection was performed by the provider based on the antibody expression levels, and the authentication was monitored in the same way. Routine mycoplasma tests were not performed for each cell line. All cell lines were kept at low passages in order to maintain their health and identity.

## Protein expression and purification for cryo-EM study

The expression construct for human Frizzled five was designed with insertion of cytochrome b562 RIL (BRIL) in the intracellular loop 3 (termed $hFzd5_{ICL3}BRIL$) with no additional thermostabilizing mutation. The chimeric protein was made using the inactive BRIL-fusion Smoothened with $V329F^{3.40}$ mutation in TM3 (PDB: ID 5L7D) as a reference, which yielded well-diffracting crystals in the report (*Byrne et al., 2016*). Wild-type hFzd5 (residue 27–428 and 441–546, UniProt ID: Q13467) was connected to BRIL using two short linkers derived from $A_{2A}$ adenosine receptor (ARRQL between residue 428 and N-terminus of BRIL, and ARSTL between C-terminus of BRIL and residue 441) and cloned with an N-terminal HA signal peptide and a FLAG epitope into the mammalian baculovirus expression (BacMam) vector pVLAD6 (*Dukkipati et al., 2008*), that has a C-terminal 3C protease site followed by a protein C epitope and an octahistidine tag. While we did not have to test multiple constructs for cryo-EM screening, we noticed that the extended helical linker was suitable for the Fab marker, which would have otherwise led to a 'collision' between the Fab/Nb unit and the detergent micelle. We also note that $hFzd5_{ICL3}BRIL$ had an overall 5-fold higher yield after expression and purification compared to hFzd5FL, indirectly suggesting its superior stability.

P0 virus was prepared in Sf9 by co-transfecting the expression vector and BestBac Linearized Baculovirus DNA (Expression Systems, CA, USA) using a standard method, and the virus was amplified from P0 to P2. $hFzd5_{ICL3}BRIL$ was expressed in HEK293S GnTI⁻ cells with BacMam baculovirus transduction. 4% (v/v) P2 virus was added to cells at a density of $2 \times 10^6$ cells per ml and culture flasks were shaken at 37 °C for 24 hr with 5% $CO_2$. After centrifugation, cells were washed with phosphate-buffered saline (PBS) supplemented with 5 mM EDTA and 1:1000 protease inhibitor cocktail (PIC, Sigma Aldrich, MO, USA), weigh and stored at −20°C. Approx. 30 grams of cell pellet was thawed and lysed with a Dounce homogenizer in a lysis buffer composed of 20 mM Tris-HCl pH 8.0, 5 mM EDTA, 2 mg/ml iodoacetamide, and 1:1000 PIC. The lysate was centrifuged at 48,000 x g for 1 hr and the membrane pellet was resuspended and nutated for 2 hr in 400 ml of a solubilization buffer consisting of 10 mM HEPES pH 7.2, 150 mM NaCl (HBS), 1% (w/v) n-dodecyl-β-D-maltopyranoside (DDM, Sol-Grade, Anatrace), 0.2% (w/v) cholesterol hemisuccinate (CHS, Anatrace), 10% (v/v) glycerol, 2 mg/ml iodoacetamide, and cOmplete PIC (Roche, Basel, Switzerland).

After centrifugation at 48,000 x g for 1 hr, 10 ml Ni-NTA resin (Qiagen, Hilden, Germany) was added to the supernatant, and the mixture was stirred at 4°C overnight. The resin was collected in a

column, washed with HBS with 0.1% (w/v) DDM, 0.02% (w/v) CHS, 10% (v/v) glycerol, and 20 mM imidazole (wash buffer) and eluted in wash buffer supplemented with 230 mM imidazole. 2 mM $CaCl_2$ was then added to the eluate, which was further purified over the in-house anti-FLAG M1 affinity sepharose column. The resin was washed with HBS + 2 mM $CaCl_2$ supplemented with detergents, which was gradually exchanged from 0.1% DDM to 0.01% GDN. The receptor was eluted in HBS with 0.002% (w/v) GDN, 0.0002% (w/v) CHS, 0.2 mg/ml FLAG peptide and 5 mM EDTA, and further purified by a size-exclusion chromatography (SEC) column Superdex 200 10/300 GL equilibrated with SEC buffer comprised of HBS, 0.002% (w/v) GDN and 0.0002% (w/v) CHS.

The anti-BRIL Fab was expressed in *E. coli* and purified as described (*Mukherjee et al., 2020*). The anti-Fab Nb (*Ereño-Orbea et al., 2018*; *Hermans et al., 2017*) was cloned in pET26b+ vector with an N-terminal hexahistidine tag followed by a TEV protease site. The protein was expressed in *E. coli* BL21(DE3) cells and purified by Ni-NTA chromatography, followed by removal of the His-tag by TEV digestion. The final step in the purification was a SEC on a Superdex 75 prep grade column whereby monodisperse Nb was obtained. The purified receptor was incubated with 1.4-fold and 2-fold molar excess of anti-BRIL Fab and anti-Fab Nb, respectively, at 4°C for overnight. The mixture was concentrated to 500 μL and injected into a SEC column Superdex 200 10/300 GL equilibrated with the SEC buffer. The peak fractions were concentrated to ~2 mg/ml and 0.01% digitonin was added at this point. The solution was further concentrated to ~8 mg/ml for cryo-EM analysis.

To prepare the Wnt/Fzd complex, we expressed XWnt8 and hFzd5FL separately from *Drosophila* S2 cells or HEK293S GnTI⁻ cells, respectively. For solubilized XWnt8, we employed the XWnt8/mFzd8CRD co-expression system developed previously (*Janda et al., 2012*). Untagged XWnt8 and C-terminal Fc fusion mFzd8CRD were stably co-expressed in cell culture media, and secreted XWnt8/mFzd8CRD-Fc was captured on Protein A sepharose resin (Sigma). After washing the resin with HBS on column, only XWnt8 was eluted from the column with Wnt elution buffer containing 10 mM Hepes-Na pH 7.2, 500 mM NaCl and 0.1% DDM, in which DDM 'wash out' lipidated Wnt from CRD with competition. The eluted XWnt8 was then captured by ConA glycan affinity resin (Vector Laboratories) for detergent exchange. The detergent composition was gradually changed as follow; 0.1% DDM, 0.09% DDM/0.01% GDN, 0.05% DDM/0.05% GDN, 0.01% DDM/0.09% GDN, 0.1% GDN and 0.01% GDN. XWnt8 in GDN is eluted with HBS containing 0.01% GDN and 250 mM each Methyl α-D-glucopyranoside and Methyl α-D-mannopyranoside (Sigma). hFzd5FL was prepared by essentially the same scheme as hFzd5$_{ICL3}$BRIL. Briefly, wild-type hFzd5FL with N-terminal FLAG and C-terminal octahistidine tags was expressed on cell membrane using BacMam system, extracted in 1% DDM/0.2% CHS and first purified with Ni-NTA resin. Pool fractions of hFzd5FL from Ni-NTA resin in 0.1% DDM/0.02% CHS was further purified with anti-FLAG M1 affinity sepharose with detergent exchange to 0.01% GDN/0.002% CHS, and a Superdex 200 10/300 GL column equilibrated with the SEC buffer. 1.2 molar excess XWnt8 was added to the SEC purified hFzd5FL, and incubated at 4°C for overnight. The mixture was concentrated to 500 μL and injected into a Superdex 200 10/300 GL column equilibrated with the SEC buffer. The peak fractions were concentrated to 5–15 mg/ml with final <1 CMC (0.06%) digitonin for cryo-EM specimen preparations.

## Cryo-EM specimen preparation and data collection

2.5 μL of either the hFzd5$_{ICL3}$BRIL/anti-BRIL Fab/anti-Fab Nb sample or the XWnt8/hFzd5FL sample was applied to glow-discharged 300 mesh gold grids (Quantifoil R1.2/1.3). Excess sample was blotted to a filter paper for 5 s before plunge-freezing using a Leica EM GP (Leica Microsystems) at 8°C and 90% humidity.

Cryo-EM data were collected at the CryoEM facility at the HHMI Janelia Research Campus. Images were collected on a Titan Krios operated at 300 kV equipped with a Gatan imaging filter and K3 camera in correlated double sampling super-resolution mode at a nominal magnification of 81,000x, corresponding to a physical pixel size of 1.078 Å. Each movie was recorded for a total of 6.125 s with 0.1225 s exposure per frame at an exposure rate of 9.5 electrons/pixel/second at the specimens, that yielded an electron counting of ~7.5 electrons/pixel/second at the camera. The datasets were collected using SerialEM (*Mastronarde, 2005*) software with the defocus range between −0.8 and −2.0 μm and beam-image shift to collect 9 movies from nine holes per stage shift and focusing.

## Cryo-EM data processing

A total 10,898 movies were collected and subjected to beam-induced motion correction using MotionCor2 (*Zheng et al., 2017*) with binning to the physical pixel size. All subsequent data processing steps were performed with cryoSPARC (*Punjani et al., 2017*). Patched contrast transfer function parameters were estimated for each micrograph from the dose-weighted averages of all frames with default parameters. Initial auto-picking templates were generated after a round of reference-free picking on a subset of micrographs. A total of 6,483,398 'particles' were picked with template-based auto picking. Several rounds of 2D classification resulted in 570,859 particles. Particles in this subgroup were subjected to several rounds of heterogeneous refinement and non-uniform refinement. A final subset of 207,021 particles were used for the final reconstruction. During the subsequent process, we noticed several 'conformations' were picked up in 3D classification attempts, as a hinge-like motion around the BRIL domain. The BRIL domain is notorious for its flexible nature, and is often poorly resolved or even disordered in crystallographic reconstructions. The final non-uniform refinement reconstruction yielded a structure at 4.6 Å resolution. The best final reconstruction (nominal resolution of 3.7 Å), was achieved with the full map fed into the 'local-refinement' routine, with a fulcrum on the BRIL domain. We attempted several ways of masking of individual parts, which did not result in a reconstruction as good as the using the full map including the detergent belt. The resolution of the final map was determined by gold standard Fourier shell correlation using the 0.143 criterion.

For the XWnt8/hFzd5FL dataset, a similar data processing approach was utilized. Initially, beam induced motion was corrected for 6,161 raw movies using MotionCor2. Dose-weighted micrographs were imported into cryoSPARC for all subsequent steps. The 'Patch CTF' routine was applied for CTF estimation, followed by template-free and template-based particle picking. A total of 6,431,867 'particles' were picked for the final set, which underwent successive rounds of 2D classification to clean up the dataset. From 2D classes alone, it was already apparent that the XWnt8/hFzd5FL was highly dynamic and did not yield any discernible secondary structure features. The resulting 245,114 particles were further processed by ab-initio reconstructions and heterogeneous refinement in an attempt to further classify subsets. The final dataset of 33,149 particles was used for the 2D classification shown in *Figure 1—figure supplement 5c* and to generate a 3D reconstruction with unsatisfying quality.

## Model building and refinement for hFzd5$_{ICL3}$BRIL/Fab/Nb

The homology model of hFzd5 was built on SWISS-MODEL server (swissmodel.expasy.org) using the crystal structures of rubredoxin fusion hFzd4TM (PDB ID: 6BD4) and BRIL fusion multi-domain Smo (PDB ID: 5L7D) as templates. The BRIL/Fab/Nb unit was modeled based on the completed Fab/Nb structure (PDB ID: 6ANI, and *Hermans et al., 2017*), and crystal structure of the BRIL/Fab complex (PDB ID: 6CBV). The hFzd5 and BRIL/Fab/Nb models are roughly fitted into the initial 4.6 Å map on UCSF Chimera. After real-space refinement with global minimization and rigid body constraints on Phenix (*Adams et al., 2010*), model was further fixed and refined iteratively using Coot (*Emsley and Cowtan, 2004*) for manual model building, inspection and correction, ISOLDE (*Croll, 2018*) on UCSF Chimera X (*Goddard et al., 2018*) for MD-based model idealization, and Phenix for real-space refinement. The map was updated to 3.9 Å then the final 3.7 Å during model building. The structure and map were visualized using UCSF Chimera X and PyMol (Schrödinger, LLC). All secondary structures in the figures were assigned using DSSP (*Joosten et al., 2011*). The cavities were detected using the CavityPlus server (*Xu et al., 2018*) with a default setting, and the dimensions of the cavities were calculated using ChExVis (*Masood et al., 2015*) server with the pocket residues assigned by CavityPlus as inputs and the probe size of 1 Å. For PDB deposition, the amino acid residues on hFzd5$_{ICL3}$BRIL were renumbered after the wwPDB validation step to separate chimeric BRIL protein from the hFzd5 region.

## STF luciferase reporter assays

STF activity assays were performed using the embryonic kidney (HEK293T) cell line, with Fzd1, 2, 4, 5, 7 and 8 knockouts. The cell line is courtesy of Michael Boutros laboratory (*Voloshanenko et al., 2017*) and stably transfected with the 7TFP plasmid (Addgene plasmid #: 24308) encoding 7xTCF-Ffluc, termed 293-Fzd-KO-STF. $1-2 \times 10^6$ cells in DMEM with 10% FBS were seeded into 6-well

plates. After 24 hrs cells were transfected with 2 µg of pRK5 plasmid containing the coding region of human hFzd5 wild-type (*Yu et al., 2012*) or mutants using Lipofectamine 2000 (Thermo Fisher) according the manufacturer's instruction. After 24 hr, cells were re-seeded in triplicate for each condition in 96-well plates, and after 20–24 hr of recovery, cells were stimulated with dilution series of recombinant Wnt3a (R and D) or Surrogate Wnt. The agonist concentrations were 0, 1.56, 6.25, 3.125 or 12.5 nM for Wnt3a and 0, 0.25, 1, 4 or 16 nM for surrogate Wnt. After 20–24 hr, cells were lysed in 30 µl 1x passive lysis buffer (Promega). 10 µl per well of lysate were assayed using the Luciferase assay kit (Promega) and luminance signal was measured using a SpectraMax plate reader. Data were analyzed and plotted in GraphPad Prism 7.

## Expression analysis of Fzd5 wild-type and mutants in 293-Fzd-KO-STF

His-tagged monomeric DRPB_Fz8 was expressed and purified as previously described (*Dang et al., 2019*) and labelled with Alexa Fluor 647 NHS (Thermo Fisher) according manufacturer's instruction. Approx. $1 \times 10^6$ 293 Fzd-KO-STF cells were transiently transfected with pRK5 plasmids encoding human hFzd5 wild-type and mutants, and plated into 96-well V-bottom plates. Assay plates were centrifuged to pellet cells, cells were washed twice in FACS buffer (PBS with 1% BSA) and stained with DRPB_Fz8 for 1 hr at room temperature. Cells were washed twice and resuspended in FACS buffer and cell surface expression of hFzd5 wild-type and mutants by means of binding of DRPB_Fz8 was determined by fluorescence intensity analysis on an Accuri C6 flow cytometer (BD Biosciences). Histogram data were plotted in GraphPad Prism 7.

## Acknowledgements

The cryo-EM data were collected at the Cryo-EM facility at the HHMI Janelia Research Campus; we thank Drs. Zhiheng Yu and Doreen Matthies for their generous supports in microscope operation and data collection. Preliminary cryo-EM screening is performed at the Cryo-EM Facility at the Department of Structural Biology, Stanford University and the Stanford-SLAC Cryo-EM facility; We thank Drs. Dong-Hua Chen and Elizabeth Montabana for their kind supports in operating the facilities and user training. Most of data were processed on the SLAC cluster; We thank Dr. Yee-Ting Li for support with computing resources. Dr. Satchal Erramilli assisted in generation of the anti-BRIL Fab. We thank Andrew Velasco for assistance, Caleb Glassman for scientific discussion.

This research was supported by NIH grants 1R01DK115728 to KCG, R01GM117372 and P50GM082545 to AAK, and by the Department of Energy, Laboratory Directed Research and Development program at SLAC National Accelerator Laboratory, under contract DE-AC02-76SF00515, to CG. KCG is an investigator of the Howard Hughes Medical Institute and the Younger Family Chair, and is supported by the Ludwig Institute. NT was supported in part by the Long-Term Fellowships from the Human Science Frontier Program Organization (LT000011/2016 L).

## Additional information

### Competing interests

Claudia Y Janda, K Christopher Garcia: KCG and CYJ are founders of Surrozen Therapeutics. The other authors declare that no competing interests exist.

### Funding

| Funder | Grant reference number | Author |
|---|---|---|
| National Institutes of Health | 1R01DK115728 | K Christopher Garcia |
| Howard Hughes Medical Institute | | K Christopher Garcia |
| Ludwig Institute for Cancer Research | | K Christopher Garcia |
| National Institutes of Health | R01GM117372 | Anthony A Kossiakoff |
| National Institutes of Health | P50GM082545 | Anthony A Kossiakoff |

| U.S. Department of Energy | DE-AC02-76SF00515 | Cornelius Gati |
| Human Frontier Science Program | LT000011/2016-L | Naotaka Tsutsumi |

The funders had no role in study design, data collection and interpretation, or the decision to submit the work for publication.

## Author contributions

Naotaka Tsutsumi, Conceptualization, Data curation, Formal analysis, Validation, Investigation, Visualization, Writing - original draft, Writing - review and editing, Cryo-EM specimen preparation, screening and data collection, Model building and refinement with KMJ and CG; Somnath Mukherjee, Investigation, Writing - review and editing, Development of anti-BRIL Fab, and purification of the Fab and anti-Fab Nb under supervision of AAK; Deepa Waghray, John S Burg, Investigation; Claudia Y Janda, Resources, Data curation, Formal analysis, Validation, Investigation, Visualization, Writing - review and editing, Mutagenesis and cell signaling assay; Kevin M Jude, Formal analysis, Validation, Writing - review and editing, Model building and inspection; Yi Miao, Investigation, Writing - review and editing, Development and purification of the surrogate Wnt agonist under supervision of KCG; Nanda Gowtham Aduri, Investigation, Preliminary cryo-EM data analysis (Krios XWnt8/ hFzd5); Anthony A Kossiakoff, Resources, Supervision, Funding acquisition, Investigation, Writing - review and editing; Cornelius Gati, Data curation, Formal analysis, Funding acquisition, Validation, Investigation, Writing - review and editing, Formal cryo-EM data analysis and map generation; Model building and inspection; K Christopher Garcia, Conceptualization, Resources, Data curation, Formal analysis, Supervision, Funding acquisition, Validation, Investigation, Writing - original draft, Project administration, Writing - review and editing

## Author ORCIDs

Naotaka Tsutsumi (iD) https://orcid.org/0000-0002-3617-7145
Somnath Mukherjee (iD) https://orcid.org/0000-0001-5447-4496
Claudia Y Janda (iD) https://orcid.org/0000-0002-3307-2559
Kevin M Jude (iD) https://orcid.org/0000-0002-3675-5136
Yi Miao (iD) https://orcid.org/0000-0003-0738-6041
Nanda Gowtham Aduri (iD) https://orcid.org/0000-0003-4301-7446
Cornelius Gati (iD) https://orcid.org/0000-0003-3693-7009
K Christopher Garcia (iD) https://orcid.org/0000-0001-9273-0278

## Decision letter and Author response

Decision letter https://doi.org/10.7554/eLife.58464.sa1
Author response https://doi.org/10.7554/eLife.58464.sa2

# Additional files

## Supplementary files

• Transparent reporting form

## Data availability

The cryo-EM map has been deposited in the Electron Microscopy Data Bank (EMDB) under accession code EMD-21927 and the model coordinate has been deposited in the Protein Data Bank (PDB) under accession number 6WW2. All cell-based assay data generated or analysed during this study are included in the manuscript and supporting files. Source data files have been provided for Figures 2 (Figure 2 - Figure supplement 1) and Figure 4 in excel format. Material availability: The expression plasmid for the anti-BRIL Fab is available from AKK (koss@bsd.uchicago.edu) by request.

The following datasets were generated:

| Author(s) | Year | Dataset title | Dataset URL | Database and Identifier |
|---|---|---|---|---|
| Tsutsumi N, Jude KM, Gati C, Garcia KC | 2020 | Structure of human Frizzled5 by fiducial-assisted cryo-EM | http://www.ebi.ac.uk/pdbe/entry/emdb/EMD-21927 | Electron Microscopy Data Bank, EMD-21927 |
| Tsutsumi N, Jude KM, Gati C, Garcia KC | 2020 | Structure of human Frizzled5 by fiducial-assisted cryo-EM | https://www.rcsb.org/structure/6WW2 | RCSB Protein Data Bank, 6WW2 |

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
