## [Decision Letter]

**Acceptance summary:**

This study reveals a new structural basis of Wnt signaling through Fzd and advances our mechanistic understanding on Wnt signaling transduction.

**Decision letter after peer review:**

Thank you for submitting your article "Structure of human Frizzled5 by fiducial-assisted cryo-EM supports a heterodimeric mechanism of canonical Wnt signaling" for consideration by *eLife*. Your article has been reviewed by two peer reviewers, and the evaluation has been overseen by a Reviewing Editor and Olga Boudker as the Senior Editor. The following individuals involved in review of your submission have agreed to reveal their identity: Rami Hannoush (Reviewer #1).

The reviewers have discussed the reviews with one another and the Reviewing Editor has drafted this decision to help you prepare a revised submission.

The reviewers have judged that your manuscript is of interest, but as described below additional re-analyses are required before it is published. In particular, please pay attention to comments 1-2 by reviewer #2, concerns were raised regarding the authenticity of the ECL structures and heterodimerization mechanism.

We would like to draw your attention to changes in our revision policy that we have made in response to COVID-19 (https://elifesciences.org/articles/57162). First, because many researchers have temporarily lost access to the labs, we will give authors as much time as they need to submit revised manuscripts. We are also offering, if you choose, to post the manuscript to bioRxiv (if it is not already there) along with this decision letter and a formal designation that the manuscript is "in revision at *eLife*". Please let us know if you would like to pursue this option. (If your work is more suitable for medRxiv, you will need to post the preprint yourself, as the mechanisms for us to do so are still in development.)

Reviewer #1:

Tsutsumi et al. present the Cryo EM structure of FZD5, which they determined with the help of a rigid BRIL fusion in intracellular loop 3 and an anti-BRIL-Fab bound by an anti-Fab nanobody to help particle alignment. The structure shows clear density for the transmembrane, the BRIL, the Fab and the nanobody domains, but not for the FZD-CRD. Another attempt at solving a structure of FZD5 in complex with XWnt8 was unsuccessful due to inherent flexibility between the FZD TMD and the CRD-Wnt complex. The TMD of FZD5 shows a very narrow allosteric binding pocket as observed previously for FZD4. Taken together the authors hypothesize that it is very unlikely that Wnt ligands exert their function via the classic allosteric binding pocket of “normal” GPCR and propose a simple heterodimerization model with LRP receptors to activate β-catenin dependent Wnt signaling. To validate this model, they compare the signaling ability of FZD5 point mutants of the allosteric pocket induced by Wnt ligands or structurally unrelated synthetic Wnt surrogates.

This is a beautiful study and the results are interesting and advance our mechanistic understanding on Wnt signaling. The manuscript is well-written, the data presented in the study are of high quality and the conclusions are supported by the data presented.

Comments:

1) It would be useful to provide some more details about the constructs design. For example, how was the insertion of the BRIL domain optimized among loops? Why not fused as continuation of appropriate transmembrane helices? Did the insertion of the BRIL domain improve the hFzd5 stability?

2) Is it possible that the insertion of the BRIL domain (or together with Fab/Nb) can affect Fzd conformation or oligomeric state (because of its bulkiness)? It is also possible that different experimental conditions result in different oligomeric states of Fzd. The authors should comment on these points.

3) The signaling strength of low-expressed/unstable Y507A is similar to (or higher than) wild-type, which is not seen in K506A. Could the author clarify this observation from the structural point of view

4) The heterodimerization model provides valuable insight into the activation mechanism of canonical Wnt signaling. The data supports that allosteric activation via small ligands of the binding pocket in the center of the TM domain does not play a major role. However, it cannot fully be excluded, and though the authors state that this cannot be fully ruled out, this point needs to be emphasized in the paper.

5) This is the first structure of a FZD TM domain without any thermostabilizing mutations. It would be informative to comment on whether any structural differences between the FZD5 and FZD4 TM can directly be attributed to such mutations.

Reviewer #2:

General assessment

Considering the fundamental importance of Wnt signaling in basic biology and medicine, as well as the scarcity of the structural information about the core signaling components, the partial structure of Fz5 reported here merits publication in *eLife*. I have no problem with the structure of TM regions and the mechanistic interpretations built upon the structural as well as functional data of the mutants. However, I found that the structural analysis and conclusion about the extracellular parts are very crude and highly misleading. Either extensive re-analyses or omission of the data/arguments is strongly advised.

Substantive concerns

1) Authenticity of the ECL structures

In contrast to the TM region of Fz5 which showed high structural similarity with that of Fz4, the extracellular region comprising three ECLs and hinge differs significantly (Figure 2C). This region is of great biological importance because it is situated in a position that could structurally and functionally link ligand-binding CRD and the TM bundle, and therefore the accuracy of the model is critical. I found that the way authors present the loop structure (Figure 1—figure supplement 4B) is highly misleading. First of all, arbitrarily separating certain chunk of EM map and overlaying the model structure should be avoided, because there is no way of knowing which part of the molecule actually contributed to the observed volume. Using the data provided, I made volume map figure myself using Chimera after normalizing the data (mean=0, stdev=1) with the threshold of 3 σ. When viewed from different orientation, the quality of the fitting becomes unacceptably low, with the region near Arg601 having almost no density. I also have serious doubts about the disulfide bonding between C190 and C590 (mislabeled as C484 in the figure). If you look closely the alignment (Figure 1—figure supplement 1), C590-containing ECL3 invariably has two Cys in all Fz/Smo except for Fz4, which are disulfide-bonded in Smo. Connecting C590 with C190 would leave the surface-exposed C592 in an unpaired state. Considering the conservation across Fzs, C190 (C232 in Fz8) is most likely bonded to preceding C182 (C228 in Fz8). Therefore, the plausible interpretation would be C182-C190 and C590-C592. This is not trivial point, because it points to the possibility that the main-chain tracing of the current model could be completely wrong, which would undermine the overall discussion about the extracellular disulfide network.

2) Heterodimerization mechanism

The authors repeatedly mention, even in the article title itself, that ligand-induced heterodimerization is the mechanism responsible for the canonical Wnt signaling. I can agree that it does not conform to the allosteric mechanism found in class A GPCR, but the choice of word (heterodimerization) will cause great confusion/misunderstanding among readers, partly because the GPCR heterodimerization has also been postulated. I understand that the authors refer to the heterodimerization between Fz5 and LRP6, and that such interaction is undeniably critical based on the numerous published results as well as the authors' own surrogate agonist experiments. But authors failed to discuss equally-well documented mechanism of Fz homodimerization, which is not mutually exclusive with the heterodimer theory. Fz CRD alone is reported to undergo homodimerization (ref 17), so is Wnt3-CRD complex (ref 10). Probably the more important question to ask is whether the Fz-LRP heterodimer undergoes homodimer(or oligomer)ization. It is regrettable that the authors did not perform more extensive analysis on the di/oligomerization tendency of FL Fz5 before and after the Wnt8 binding. I would have liked it very much if statistical analysis on the dimer/monomer ratio found in the earlier SEC fractions were presented.

[Editors' note: further revisions were suggested prior to acceptance, as described below.]

Thank you for submitting your article "Structure of human Frizzled5 by fiducial-assisted cryo-EM supports a heterodimeric mechanism of canonical Wnt signaling" for consideration by *eLife*. Your article has been reviewed by two peer reviewers, and the evaluation has been overseen by a Reviewing Editor and Olga Boudker as the Senior Editor. The following individuals involved in review of your submission have agreed to reveal their identity: Rami Hannoush (Reviewer #1); Junichi Takagi (Reviewer #2).

In light of their advice I am delighted to say that we are happy, in principle, to publish a suitably revised version in *eLife*.

Please note that reviewer #2 commented on your revision and suggested a textual change in Abstract regarding the word "ligand-induced heterodimerization” and making appropriate adjustment in both Figure 1E and Figure 1—figure supplement 5E.

We therefore invite you to revise your paper one last time to edit your manuscript to comply with our format requirements and to maximise the accessibility and therefore the impact of your work.

Reviewer #1:

The authors have fully addressed my comments and this is an improved version of the paper that is ready for publication.

Reviewer #2:

The authors addressed the primary concern I expressed in the initial review (i.e., the authenticity of the ECL structure) by removing some segments from the model and accordingly modifying the figures. Overall, I think the paper is almost ready for the publication in *eLife* considering the fundamental importance of Wnt signaling in basic biology and medicine, as well as the scarcity of the structural information about the core signaling components. However, I could not find satisfying changes in the text regarding my second concern (i.e., clarification of the terminology of "heterodimerization mechanism").

1) Heterodimerization mechanism

Maybe my comment in the previous review was not clear. I was not suggesting the authors to emphasize the alternative signaling mechanism involving Fz-Fz homodimerization. My concern was the choice of the word (heterodimerization mechanism) because readers may get the wrong impression that they are claiming the GPCR (Fz) heterodimerization as the major signaling mechanism, particularly when readers only read Abstract. In the Abstract, I suggest changing "ligand-induced heterodimerization" to something like "Fzd/LRP6 heterodimerization".

2) EM images of Wnt8-Fz5

The authors revised the text to further emphasize the limitation of the data, which is fair and appropriate. And I can agree with their intention to keep the figure as the main figure. However, the newly added 3D cross-section of the Figure 1E is highly inappropriate. First of all, the 2D class averages of the original Figure 1E and Figure 1—figure supplement 5 are of low quality and can contain many misaligned particles. In general, calculating a 3D volume map does not make the data more reliable than calculating 2D averages (it is the opposite). More importantly, the "local refinement" feature in cryoSPARC program can only be performed after accurately determining the overall 3D structure, which is not the case here. This point is clearly mentioned in the developer's website (https://cryosparc.com/blog/local-refinement-snrnp-case-study) as "Before we can use any of the local refinement tools, we need an accurate picture of the overall structure…". Besides, there are no description about the mask they used in the local refinement nor the explanation of the heat map values. I strongly suggest authors to remove the "refined 3D map" for the Wnt-Fz complex from both Figure 1E and Figure 1—figure supplement 5E.

---

## [Author Response]

Reviewer #1:Comments:1) It would be useful to provide some more details about the constructs design. For example, how was the insertion of the BRIL domain optimized among loops? Why not fused as continuation of appropriate transmembrane helices? Did the insertion of the BRIL domain improve the hFzd5 stability?

As we elaborate in the Materials and methods Section, the BRIL domain is connected to TM5 and TM6 of hFzd5 by two short peptides (ARRQL and ARSTL) derived from TM5 and TM6 of A_2A_ adenosine receptor. TM5/6 of hFzd5, α-helix 1/4 of BRIL, and the two peptides formed continuous α-helices that permit cryo-EM particle alignment. In this study, we have not compared the stability of hFzd5_ICL3_BRIL and hFzd5FL, for instance, by assessing the thermostability. However, we have noticed an overall 5-fold higher yield of the hFzd5_ICL3_BRIL construct after expression and purification compared to hFzd5FL, suggesting superior stability. We have edited the Materials and methods section of the manuscript to address these points.

2) Is it possible that the insertion of the BRIL domain (or together with Fab/Nb) can affect Fzd conformation or oligomeric state (because of its bulkiness)? It is also possible that different experimental conditions result in different oligomeric states of Fzd. The authors should comment on these points.

We agree that we cannot entirely exclude that the BRIL insertion into the ICL3 of Fzd5 alters the propensity of Fzd5 for distinct conformational and oligomeric states. In reality, cryo-EM analysis and protein crystallization techniques both have aspects that can bias towards certain conformational states, but we are careful not to extend our interpretations beyond the most plausible explanations.

Concerning the oligomeric state, we predominantly observed peaks of monomeric hFzd5I_CL3_BRIL and hFzd5FL on size-exclusion chromatography. The BRIL insertion does not noticeably change the preference for distinct oligomeric states. Also, to our knowledge, it has not been reported that the insertion of BRIL into the ICL3 of GPCRs triggers changes to their oligomeric states. Hence, we focus our discussion on the description of the oligomeric states without extending our speculations on their functional implications.

3) The signaling strength of low-expressed/unstable Y507A is similar to (or higher than) wild-type, which is not seen in K506A. Could the author clarify this observation from the structural point of view

We thank the reviewer for this comment. Our flow cytometry data show that the K506A variant has absolutely no detectable expression, whereas the Y507A variant has a relatively low cell surface expression that was enough to saturate the Wnt3a-dependent signaling. This suggests while the Y507A mutation impairs expression, the K506A is detrimental to the stability of hFzd5. Among ten Frizzled family members, K^7.41^ is 100% conserved where the sidechain plays a role in the inter-helical packing between TM6 and TM7 as mentioned, by bridging Y^6.51^ (90% conserved, L for Fzd8), E^6.54^ (100% conserved) and Y^7.42^ (70% conserved, I for Fzd4/9/10) in the structure, which suggests its functional importance. We edited the corresponding discussion lightly to imply the class-conserved K506 is more important than Y507.

4) The heterodimerization model provides valuable insight into the activation mechanism of canonical Wnt signaling. The data supports that allosteric activation via small ligands of the binding pocket in the center of the TM domain does not play a major role. However, it cannot fully be excluded, and though the authors state that this cannot be fully ruled out, this point needs to be emphasized in the paper.

We agree that although the close packing of the TMs suggests that the pocket of the receptor occludes small molecules, we cannot fully exclude that certain classes of small molecules can be accommodated in the “ligand-binding” pocket. In fact, we have cited the publication reporting the Smo agonist SAG1.3 could act as a partial agonist for Fzd6-Gi signaling. This is in good agreement with our existing discussion regarding the mechanism of Fzd receptor activation.

Our data suggest that simple cross-linking of Fzd and LRP6 can trigger the intracellular β-catenin signaling, as the surrogate agonists have shown. This mechanism is not mutually exclusive with alternative mechanisms to activate intracellular signaling pathways that might be triggered by small-molecule ligands. Hence, we further interpret that the conformational change model might be more applicable to intracellular signaling mediated by G proteins. Fzd-dependent G protein signaling might be more sensitive to conformational change. The possibility of conformational change during G protein signaling has been included in the current revised version.

5) This is the first structure of a FZD TM domain without any thermostabilizing mutations. It would be informative to comment on whether any structural differences between the FZD5 and FZD4 TM can directly be attributed to such mutations.

The stabilizing mutations of Fzd4 were designed by the group of Prof. Xu based on multiple sequence alignment of Fzd members. For three of the four mutations, amino acids that occur naturally in Fzd5 were chosen (C450I, C507F, and S508Y for the Fzd4 structure). The remaining one is also similar (M309L for the Fzd4 structure, and Fzd5 has V at the corresponding position). Based on this analysis, we hypothesize that Fzd5 is perhaps naturally more stable compared to Fzd4. Given these similarities, we cannot draw any meaningful conclusion by comparing variations at these particular sites.

Reviewer #2:Substantive concerns1) Authenticity of the ECL structuresIn contrast to the TM region of Fz5 which showed high structural similarity with that of Fz4, the extracellular region comprising three ECLs and hinge differs significantly (Figure 2C). This region is of great biological importance because it is situated in a position that could structurally and functionally link ligand-binding CRD and the TM bundle, and therefore the accuracy of the model is critical. I found that the way authors present the loop structure (Figure 1—figure supplement 4B) is highly misleading. First of all, arbitrarily separating certain chunk of EM map and overlaying the model structure should be avoided, because there is no way of knowing which part of the molecule actually contributed to the observed volume. Using the data provided, I made volume map figure myself using Chimera after normalizing the data (mean=0, stdev=1) with the threshold of 3 σ. When viewed from different orientation, the quality of the fitting becomes unacceptably low, with the region near Arg601 having almost no density. I also have serious doubts about the disulfide bonding between C190 and C590 (mislabeled as C484 in the figure). If you look closely the alignment (Figure 1—figure supplement 1), C590-containing ECL3 invariably has two Cys in all Fz/Smo except for Fz4, which are disulfide-bonded in Smo. Connecting C590 with C190 would leave the surface-exposed C592 in an unpaired state. Considering the conservation across Fzs, C190 (C232 in Fz8) is most likely bonded to preceding C182 (C228 in Fz8). Therefore, the plausible interpretation would be C182-C190 and C590-C592. This is not trivial point, because it points to the possibility that the main-chain tracing of the current model could be completely wrong, which would undermine the overall discussion about the extracellular disulfide network.

We agree with the reviewer that our initial model was not sufficiently conservative. Our interpretation of the EM volume was C182(unmodeled)-C486/C190-C484, however, we also made variants with C182(unmodeled)-C190/C484-C486 during our model building attempts, and we agree that it is not appropriate to judge the disulfide bonding network based on this map.

Based on the reviewer’s suggestions, we have made the following adjustments.

1) We have removed residues P487 to P498, which includes the flexible region in ICL3 and R495, from our model. We agree that the EM volume of this region is ambiguous. Although ICL3 likely adopts multiple conformations, including the one we have previously modeled, we recognize that it implies false precision when a single model is placed (subsection “Structural comparison between hFzd5, hFzd4 and hSmo”).

2) We have removed residues F188 to C192 and the sidechain of C484 from our model. This is to avoid misleading conclusions about the most flexible yet important regions of the structure. Further, we have revised the Discussion in our manuscript accordingly. We note that the hinge domain and ICL3 of the Fzd5/8 subfamily have a unique distribution of cysteine residues among the class F GPCRs, and wish to perform additional experiments to more adequately address this question.

Further, we apologize for the inconsistency in residues numbering. In the initial submission, we provided the coordinates directly from Phenix refine. Here, we have attached a revised model with renumbered residues, so it is consistent with the figures.

2) Heterodimerization mechanismThe authors repeatedly mention, even in the article title itself, that ligand-induced heterodimerization is the mechanism responsible for the canonical Wnt signaling. I can agree that it does not conform to the allosteric mechanism found in class A GPCR, but the choice of word (heterodimerization) will cause great confusion/misunderstanding among readers, partly because the GPCR heterodimerization has also been postulated. I understand that the authors refer to the heterodimerization between Fz5 and LRP6, and that such interaction is undeniably critical based on the numerous published results as well as the authors' own surrogate agonist experiments. But authors failed to discuss equally-well documented mechanism of Fz homodimerization, which is not mutually exclusive with the heterodimer theory. Fz CRD alone is reported to undergo homodimerization (ref 17), so is Wnt3-CRD complex (ref 10). Probably the more important question to ask is whether the Fz-LRP heterodimer undergoes homodimer(or oligomer)ization. It is regrettable that the authors did not perform more extensive analysis on the di/oligomerization tendency of FL Fz5 before and after the Wnt8 binding. I would have liked it very much if statistical analysis on the dimer/monomer ratio found in the earlier SEC fractions were presented.

In several crystal structures, the isolated Fzd-CRDs in complex with lipids or Wnt3a were observed in dimeric arrangements, and we have cited the corresponding publications. However, we have previously carried out single-molecule TIRF microscopy imaging of full-length Fzd8 and LRP6 on live cell membranes with and without Wnt (Janda et al., 2017). We have found that the predominant state of the signaling complexes is a 1:1 monomer of the Fzd8/LRP6 heterodimer, with less populated higher oligomeric states. We have not seen evidence of Fzd homodimers on cells, although this issue remains to be fully resolved. We have also observed and described a small population of hFzd5_ICL3_BRIL/Fab/Nb dimers in the cryo-EM 2D class averages. Given the published data in hand, we feel that we have given a fair and transparent account of this issue and do not think that anything else is within the scope of the current paper.

We also note that purified hFzd5FL and hFzd5_ICL3_BRIL in detergent micelles elute from size-exclusion chromatography (SEC) at volumes predictive of monomers. The addition of XWnt8 only leads to a marginal faster migration through the size-exclusion column, suggesting a 1:1 complex formation. However, the SEC profiles also show some higher molecular weight shoulders. These are very common for most other GPCRs we have purified, and the amount of the small shoulder varies between experiments and protein preparations. As we cannot know whether it represents the physiological dimer (multimer) or artifacts (such as misfolded and aggregated receptors, and unnatural head-to-tail dimers often observed for detergent-solubilized membrane proteins), we are hesitant to quantify the ratio based on the unclear SEC observation that can lead to erroneous conclusions.

[Editors' note: further revisions were suggested prior to acceptance, as described below.]

Reviewer #2:The authors addressed the primary concern I expressed in the initial review (i.e., the authenticity of the ECL structure) by removing some segments from the model and accordingly modifying the figures. Overall, I think the paper is almost ready for the publication in eLife considering the fundamental importance of Wnt signaling in basic biology and medicine, as well as the scarcity of the structural information about the core signaling components. However, I could not find satisfying changes in the text regarding my second concern (i.e., clarification of the terminology of "heterodimerization mechanism").1) Heterodimerization mechanismMaybe my comment in the previous review was not clear. I was not suggesting the authors to emphasize the alternative signaling mechanism involving Fz-Fz homodimerization. My concern was the choice of the word (heterodimerization mechanism) because readers may get the wrong impression that they are claiming the GPCR (Fz) heterodimerization as the major signaling mechanism, particularly when readers only read Abstract. In the Abstract, I suggest changing "ligand-induced heterodimerization" to something like "Fzd/LRP6 heterodimerization".

As suggested by the reviewer, we revised the word in the Abstract and the impact statement.

2) EM images of Wnt8-Fz5The authors revised the text to further emphasize the limitation of the data, which is fair and appropriate. And I can agree with their intention to keep the figure as the main figure. However, the newly added 3D cross-section of the Figure 1E is highly inappropriate. First of all, the 2D class averages of the original Figure 1E and Figure 1—figure supplement 5 are of low quality and can contain many misaligned particles. In general, calculating a 3D volume map does not make the data more reliable than calculating 2D averages (it is the opposite). More importantly, the "local refinement" feature in cryoSPARC program can only be performed after accurately determining the overall 3D structure, which is not the case here. This point is clearly mentioned in the developer's website (https://cryosparc.com/blog/local-refinement-snrnp-case-study) as "Before we can use any of the local refinement tools, we need an accurate picture of the overall structure…". Besides, there are no description about the mask they used in the local refinement nor the explanation of the heat map values. I strongly suggest authors to remove the "refined 3D map" for the Wnt-Fz complex from both Figure 1E and Figure 1—figure supplement 5E.

As suggested by the reviewer, we deleted all data regarding the attempts at 3D refinement of XWnt8/hFz5FL.